# A Cohort Study of Korean Radiation Workers: Baseline Characteristics of Participants

**DOI:** 10.3390/ijerph17072328

**Published:** 2020-03-30

**Authors:** Soojin Park, Songwon Seo, Dalnim Lee, Sunhoo Park, Young Woo Jin

**Affiliations:** Laboratory of Low Dose Risk Assessment, National Radiation Emergency Medical Center, Korea Institute of Radiological and Medical Sciences, 75 Nowon-ro, Nowon-gu, Seoul 01812, Korea

**Keywords:** occupational exposure, neoplasm, radiation, epidemiology

## Abstract

The Korean Radiation Worker Study investigated the health effects of protracted low-dose radiation among nuclear-related occupations in the Nuclear Safety and Security Commission in Korea. From 2016–2017, 20,608 workers were enrolled (86.5% men and 30.7% nuclear power plant workers). The mean cumulative dose ± standard deviation between 1984 and 2017 (1st quarter) was 11.8 ± 28.8 (range 0–417) mSv. Doses below recording level (≤0.1 mSv) were reported in 7901 (38.3%) cases; 431 (2%) had cumulative doses ≥100 mSv. From 1999–2016, 212 cancers (189 men, 23 women) occurred; thyroid cancer predominated (39.2%, 72 men, 11 women). In men, the standardized incidence ratio (SIR) for all cancers was significantly decreased (SIR = 0.76, 95% CI 0.66–0.88); however, that for thyroid cancer was significantly increased (SIR = 1.94, 95% CI 1.54–2.44). Compared to the non-exposed group (≤0.1 mSv), the relative risk (RR) in the exposed group (>0.1 mSv) after adjusting for sex, attained age, smoking status, and duration of employment was 0.82 (95% CI 0.60–1.12) for all cancers and 0.83 (95% CI 0.49–1.83) for thyroid cancer. The preliminary findings from this baseline study with a shorter follow-up than the latency period for solid cancer cannot exclude possible associations between radiation doses and cancer risk.

## 1. Introduction

The current radiation protection standards, including the dose limits for the public and radiation workers, are mainly based on studies of the Japanese atomic bomb survivors who experienced acute exposure to a broad range of radiation doses. By contrast, radiation workers are typically exposed to protracted low doses and low-dose rates during their employment. Thus, studies on radiation workers can provide more practical evidence of the health risks of low-dose radiation exposure in our daily lives, although such studies are often plagued by selection bias.

The recent large International Nuclear Workers Study (INWORKS) of radiation workers indicated that the dose–risk association for workers is consistent with that for the Japanese atomic bomb survivors; however, the risk remains unclear at low-dose ranges [1,2,3]. Various studies of radiation workers in different countries have indicated lower rates for disease incidence or mortality than in general populations; these findings could be interpreted as the healthy worker effect, which is a type of selection bias, typically seen in occupational epidemiology; however, dose–response relationships in these studies were variable, and mostly not statistically significant [4,5,6]. In addition to the heterogeneity of population characteristics, including baseline health status across countries, these inconsistent and limited findings are mainly attributable to a limited sample size and/or follow-up period, and a lack of confounding information such as lifestyle factors and socioeconomic status. In addition, a few studies have been conducted among nuclear-related workers in South Korea [7,8]. However, due to sparse information on confounding factors and a short follow-up period, the study findings were limited. Moreover, additional follow-up information pertaining to the participants of these studies is no longer available owing to the reformed Personal Protection Act [9].

In order to overcome the limitations of the previous studies, we launched a national cohort study of radiation workers (aka “the Korean Radiation Worker Study (KRWS)”) in various nuclear-related occupations for long-term follow-up, and enrolled study participants to the cohort through a nationwide baseline survey from 2016–2017 [10]. We collected information including demographics, occupational characteristics, and lifestyle factors of the workers, and linked the data with the national dose registry and the national cancer registry. As this is the first study after cohort enrollment, we intended to report baseline results, including occupational characteristics and cancer incidence derived from the cohort. 

## 2. Materials and Methods

### 2.1. Study Population and Cohort Enrollment

The design and methods of the KRWS were previously described in its protocol [10]. In summary, a nationwide self-administered survey of demographics, lifestyle factors, and work practices was conducted on 42,607 Korean radiation workers from 24 May, 2016 to 30 June, 2017 at educational institutions where all radiation workers receive radiation safety education every year; among them, 35,789 workers participated in the survey. After collecting the names and dates of birth of the participants through the surveys, we provided this information to the organization (i.e., Korea Foundation of Nuclear Safety) in charge of the national dose registry for linking radiation doses. We then requested another organization (i.e., Korea National Cancer Center) in charge of the national cancer registry to link the cancer status of all subjects identified from the dose registry. After excluding subjects who responded more than once, or who would not be able to complete follow-up due to unidentified personal identification numbers or disagreement with study participation, a total of 20,608 workers were enrolled in the cohort (Figure 1).

### 2.2. Dosimetry and Data for Cancer Incidence

The sources of dosimetry data and cancer incidence data have been described in detail in the study protocol [10]. The radiation doses of the workers collected in this study were personal dose equivalent (*H*_p_(10)) from personal badge dosimeters reported to the Central Registry for Radiation Worker Information (CRRWI) from 1984 to the first quarter of 2017. Radiation doses ≤0.1 mSv were recorded as “below recording level” and were considered as a dose of zero for external exposure. The CRRWI has information on occupational classifications, including nuclear power plants, industrial radiography, industry, including production and sales, medical institutes (mostly nuclear medicine and radiotherapy), education institutes, research institutes, and public institutes, and this information is linked to radiation doses for individual workers. Workers who work simultaneously in two or more institutions wear a personal badge dosimeter for each institution and report their radiation doses separately. The reported doses are then integrated by the worker’s personal identification number to manage exposure levels for radiation protection.

The cancer incidence status of individual workers was identified using the International Classification of Diseases (ICD)-10 codes from the National Cancer Registry for the period of 1 January 1999 to 31 December 2016.

### 2.3. Data Analysis

We linked the data from the survey with personal radiation doses and the cancer registry via personal identification numbers. Demographics and characteristics of the work practices of the enrolled cohort were summarized using descriptive statistics. For external comparisons of cancer incidence in the cohort with that in the Korean general population, age (5-year interval)- and sex-specific standardized incidence ratios (SIR) and 95% confidence intervals (CI) were calculated. Since the cancer statistics of the general population were available from 1999, the start of follow-up was set to the date of first employment or 1 January 1999, whichever was later. The exit of follow-up was set to the date of the cancer diagnosis or 31 December 2016, whichever was earlier. When a person had multiple cancers, the date of cancer diagnosis was based on the first cancer. For the internal comparison, we estimated relative risks (RR) with Poisson regression, comparing cancer incidence rates between the exposed group and the non-exposed group (defined as workers whose cumulative radiation doses were “below recording level,” ≤0.1mSv) with and without adjustment for sex, attained age, smoking status, and duration of employment. Subjects who had been diagnosed with cancer prior to their employment (n = 83), those diagnosed before 1999 (n = 8), and those who started work after 31 December 2016 (n = 990) were excluded from the cancer-related analyses; finally, 19,527 workers were included in this analytic sample to examine cancer risk at this early stage of the cohort study. Owing to the exploratory nature of the analyses at the early stage of the study, multiplicity adjustments were not made. Statistical analyses were conducted using EPICURE software (Risk Sciences International, Version 2.0, Ottawa, Ontario, Canada).

### 2.4. Ethics Approval

All study participants provided informed written consent prior to study enrollment, and this study has received ethical approval from the institutional review board of the Korea Institute of Radiological and Medical Sciences (IRB No.K-1603-002-034). The investigations were performed following the rules of the Declaration of Helsinki of 1975, revised in 2013.

## 3. Results

### 3.1. Baseline Characteristics of the Cohort

A total of 20,608 radiation workers were enrolled in the cohort; this accounted for approximately 50% of all workers registered with the CRRWI in 2017. The demographic characteristics and occupational history of the cohort are summarized in Table 1 and Table 2. The majority of the cohort comprised men (86.5%), and nearly half of the workers were born after 1981. Among the eight facility-based occupations, which formerly included 10 occupations and a few similar occupations such as industry, production, and sales recently combined, nuclear power plant workers comprised the majority, followed by industry and industrial radiography workers. Less than 10% of the cohort had received warnings for exceeding 5 mSv per quarter or had abnormal white blood cell counts during their employment periods. The baseline characteristics by occupation are provided in Appendix A.

### 3.2. Distribution of the Radiation Dose of the Cohort

Overall, the annual average doses for the cohort gradually decreased over time from 4.5 mSv in 1984 to 0.6 mSv in 2016 (Table 3). The proportion of workers receiving doses >20 mSv continued to be less than 1% since the late 1990s (Table 3). Among the eight occupations, the annual average radiation dose of industrial radiography was higher than that of other occupations over time (Figure 2). The overall annual average radiation dose of industrial radiography was 2.69 mSv, followed by nuclear power plant workers at 1.56 mSv, medical institute workers at 1.03 mSv, and other occupations at less than l mSv.

The distribution of the cumulative dose in the cohort skewed to the right (Figure 3). This is a typical distribution shown in most radiation epidemiological studies [11,12,13], with a mean cumulative dose ± standard deviation of 11.8 ± 28.8 mSv (range 0–417 mSv) and a median cumulative dose of 0.59 mSv (interquartile range 0–9.1 mSv). Radiation doses below recording level (i.e., ≤0.1 mSv, considered as a cumulative dose of zero) were recorded in 7901 (38.3%) workers; 431 workers had a cumulative dose of ≥100 mSv, corresponding to approximately 2% of the cohort.

### 3.3. Cancer Incidence of the Cohort

Among 19,527 subjects available for analyses, a total of 212 cancer cases (189 in men and 23 in women) were identified from 1999–2016, and the total follow-up was 158,815.38 person-years with a mean of 8.13 (± 6.29) years. Among the cancer cases, thyroid cancer was the most prevalent (39.2%), followed by stomach cancer (21.2%) and colon cancer (7.1%). There were two leukemia cases (0.9%).

Table 4 and Table 5 demonstrate the external comparisons of cancer incidence using SIRs. Overall, the SIRs for all cancers combined were decreased in both, men (SIR = 0.76, 95% CI 0.66–0.88) and women (SIR = 0.84, 95% CI 0.56–1.26) in our cohort; however, statistical significance was found only in men (Table 4). For the individual cancer sites, significant decreases in SIRs were observed for liver cancer (SIR = 0.20, 95% CI 0.10–0.43) and lung cancer (SIR = 0.23, 95% CI 0.09–0.61) in men. A statistically significant increase in SIR was observed for thyroid cancer in men (SIR = 1.94, 95% CI 1.54–2.44) and for cancer of the uterus, part unspecified, in women (SIR = 25.86, 95% CI 3.64–183.60).

Age- and sex-specific SIRs for all cancers combined, all cancers combined excluding thyroid cancer, and thyroid cancer stratified by occupations and birth year, are shown in Table 5. Overall, the SIR for all cancers combined tended to be low for all occupations except those related to public institutes and the military, and statistical significance was found for industrial radiography (SIR = 0.36, 95% CI 0.20–0.63) and medical institute (SIR = 0.63, 95% CI 0.43–0.93). Conversely, the SIR for thyroid cancer tended to be high for all occupations except those related to education institutes and industrial radiography, and statistical significance was found for occupations related to nuclear power plants (SIR = 2.68, 95% CI 1.97–3.65) and the military (SIR = 8.26, 95% CI 2.07–33.04). The SIR for thyroid cancer tended to increase in all groups of subjects born since the 1950s.

In order to examine cancer incidence associated with radiation exposure, age- and sex-specific SIRs for the selected cancers (i.e., all cancers combined, all cancers combined excluding thyroid cancer, and thyroid cancer) were estimated in the exposed group and the non-exposed group (≤0.1 mSv), and RRs were also estimated to compare cancer risk between the two groups. The characteristics including person-years and demographics of each group are presented in Appendix A. The SIRs for these cancers were not significantly different between the two groups, and the RRs were not statistically significant after adjusting for sex, attained age, birth year, smoking status, and duration of employment (Table 6).

## 4. Discussion

The present study describes the baseline characteristics, including demographics, occupational history, and cancer incidence, of the cohort of radiation workers in Korea. While previous studies of radiation workers in Korea mainly targeted nuclear power plant workers or medical diagnostic workers [7,8,14,15], this cohort included all nuclear-related occupations, enabling an evaluation of radiation-induced health effects of radiation workers in various nuclear facilities, including industrial radiographers whose annual radiation doses were the highest among all occupations [16]. In particular, in the baseline survey for this study, industrial radiographers were also exposed to relatively high potential health risks such as night shifts, smoking, and high BMI; therefore, their radiation doses and health statuses should be monitored more carefully.

Radiation doses (i.e., personal dose equivalent (*H*_p_(10)) of the cohort steadily decreased over the years, and the annual average dose has been close to or below 1 mSv for the last 10 years. This trend was also observed worldwide [17,18] and among Korean diagnostic radiation workers who were not included in this cohort [19]. The mean cumulative dose of the cohort over the period 1984–2017 (1st quarter) was 11.8 mSv, which increased from 6.1 mSv over the period 1984–2004 of the previous retrospective cohort study on Korean radiation workers [7], and is lower than that of the INWORKS, which reported a mean dose of 25 mSv during 1945–2005 [1].

In addition to radiation exposure in workplaces, radiation workers can be exposed to radiation from medical imaging and treatments. Such medical radiation exposure has increased rapidly, especially in health care level I countries including South Korea [20]. In our cohort, approximately 86% of subjects received x-ray imaging and approximately 25% underwent computed tomography (CT) in the last three years (Appendix A). These values may surpass that of the general population and office workers (e.g., general affairs, administration, teachers) in Korea, considering that non-office workers (e.g., radiation workers, health care providers, drivers) receive annual health examination opportunities in the national health examination system, whereas regional policyholders and office workers receive them on a biennial basis [21,22]. In addition, examination participation (i.e., health checkups) rates of non-office workers were higher than those of regional policyholders and office workers [23]. Compared with occupational exposure, the recent annual levels of which steadily decreased around dose limits for the public (i.e., 1 mSv), levels of medical exposure are not negligible, implying that it is necessary to consider medical exposure in estimating radiation-induced health risks.

We found that the incidences of liver and lung cancers in this cohort were significantly lower than those of the general population. Moreover, a tendency of a low SIR for all cancers combined, excluding thyroid cancer, was observed for all occupation types and in both, the exposed and non-exposed groups, implying a healthy worker effect [24,25]. This is a typical phenomenon in occupational cohort studies, and other studies also reported low cancer incidences or mortality among radiation workers [4,26,27,28,29,30,31]. In the present study, our cohort included active workers during the follow-up period for cancer incidence, and therefore they were presumed to mostly have a good health status. Indeed, workers born before the 1950s, which is the oldest birth cohort in this study, had the lowest SIR for all cancers combined.

The SIR for thyroid cancer in this cohort was significantly high (age-and sex-specific SIR = 1.69, 95% CI 1.36–2.10); this finding is comparable to those of other previous studies in Canada (SIR = 1.32, 95% CI 0.97–1.75) [32], the United States (SIR = 2.23, 95% CI=1.29–3.59) [33], and Korea (SIR for radiologic technologists = 2.14, 95% CI 1.29–3.35; SIR for nuclear power plant workers = 5.93, 95% CI 2.84–10.90) [8,14]. It is well known that cancer screening related to high levels of access to healthcare could play a major role in the increase in thyroid cancer incidence [34,35,36]. Indeed, the SIRs for thyroid cancer tended to be low among workers in industrial radiography and education institutes in the present cohort, and their thyroid screening rates (6% for the industrial radiography and 14 % for education institutes) were also lower than that of other occupations (Appendix A). The highest SIR for thyroid cancer was observed in the military, where two cases occurred among 164 workers. Since the cumulative radiation doses in both cases were <1 mSv, the high SIR may not be related to their occupational exposure. Moreover, since cancer risks between the exposure and non-exposure groups were not different in the present study, the current high SIR for thyroid cancer in this cohort may be mainly attributed to factors other than occupational exposure. However, as the thyroid is highly sensitive to radiation and some studies indicated a possible association between occupational exposure and a high incidence of thyroid cancer [37,38,39], and since the current study provides preliminary findings from the baseline study of our cohort with a limited follow-up, we cannot exclude a possible association between radiation doses and cancer risk. Further analyses for dose-response are necessary to shed light on radiation-induced cancer risk.

In addition to thyroid cancer, a high SIR for cancer of the uterus, part unspecified, was observed in women. Since the cumulative radiation dose of the case was zero, this high SIR was not considered to be related to occupational exposure. However, as this cancer is rare with an incidence rate of approximately one per 100,000 persons in the Korean general population in the age corresponding to the case, continued monitoring is needed for ensuring occupational health safety among radiation workers.

As cohort enrollment was conducted through a survey of active workers, this cohort had some limitations. First, our subjects may not accurately represent the target population. Although we conducted 602 nationwide surveys to recruit subjects to the cohort, that covered approximately 95% of the mandatory radiation safety education for radiation workers for approximately a year, we cannot rule out selection bias. However, selection bias and low participation in a cohort study are not likely to substantially influence exposure–disease association [40,41], although occupational epidemiology is often plagued by the healthy worker effect, which is a special type of selection bias. Given that the distributions of the number of workers and radiation doses by occupation types in the cohort did not deviate considerably from the target population [16], and workers (i.e., target population) were in active service with mostly healthy status when they responded to the survey, the selection of study subjects was not expected to be strongly associated with the exposure and disease status. Second, our cohort was relatively young (with a mean age of 38.3 years and a mean follow-up of 8.1 years), when compared with the recent multinational international cohort study (INWORKS) (that has a mean age of 58 years and a mean follow-up of 27.0 years [42]). Moreover, considering that the usual latency period for solid cancer is at least 5–10 years, which is longer than the average follow-up in this study, this cohort has limited statistical power at this stage, and continued long-term follow up is required to extract full value from the cohort. However, a younger cohort provides some advantages in terms of having fewer uncertainties about information collected such as dosimetry and baseline health status. For instance, older workers with long-term careers may have greater uncertainties about their radiation doses than recently hired workers, due to changes in personal dosimeter types (e.g., film badge or thermoluminescent dosimeters) over long employment periods and deficient or missing dose records (e.g., undocumented dose records) from before 1984, the year from when the national dose registry was available in Korea [10]. In addition, a younger cohort is assumed to be healthy; this is more appropriate for study subjects using a prospective cohort design. Third, the size of our cohort was smaller than cohorts from other countries such as the United Kingdom, the United States, and France [43,44,45,46]. In order to increase the cohort size, enrollment will be continued at five-year intervals, and the development of a strategy to enroll retired workers is now in progress [10]. Lastly, the current occupational classification in the dose registry does not clearly reflect the occupational nature (e.g., profession). In particular, public institutes and the military, where workers are not classified based on job characteristics, may include various occupational types such as research, education, and industry. Therefore, it is necessary to obtain task-related information to characterize the profession of study subjects.

Despite these limitations, this cohort is unique as it has been designed for a prospective cohort study covering all nuclear-related occupation types. We also collected information on factors such as occupational history (e.g., night shift, employment status, and radiation source) and lifestyle factors (e.g., smoking, alcohol, and BMI) for baseline characteristics of the cohort, and this information was linked to the national registries for radiation doses and cancer. The completeness of the cancer registry in Korea was estimated to be about 98% [47]; this may minimize potential misclassification bias for cancer diagnosis. We will further update radiation doses including internal doses, and estimate organ absorbed doses, as the current radiation doses collected were not physical quantities but radiation protection quantities not appropriate for risk assessment of individual organs or tissues; it is also necessary to collect data regarding other health outcomes such as non-cancer diseases and laboratory biomarkers from the National Health Insurance Sharing Service database [10].

## 5. Conclusions

In conclusion, we have established a prospective cohort of radiation workers covering all nuclear-related occupations in Korea. This cohort provides comprehensive individual information including occupational and demographic characteristics, obtained through a nationwide survey, radiation dosimetry, and cancer incidence. The baseline findings did not deviate from those of other studies, indicating a healthy worker effect for all cancers combined and an increase in thyroid cancer incidence compared to the general population. As this cohort is relatively young with limited follow-up, continued follow-up is needed to investigate radiation-induced health risks. Moreover, the cohort should be expanded to include retired workers; this would allow more precise quantification of the dose–response relationship. Further studies on non-cancer diseases are also being planned with linkage to the National Health Insurance database.

## Figures and Tables

**Figure 1 ijerph-17-02328-f001:**
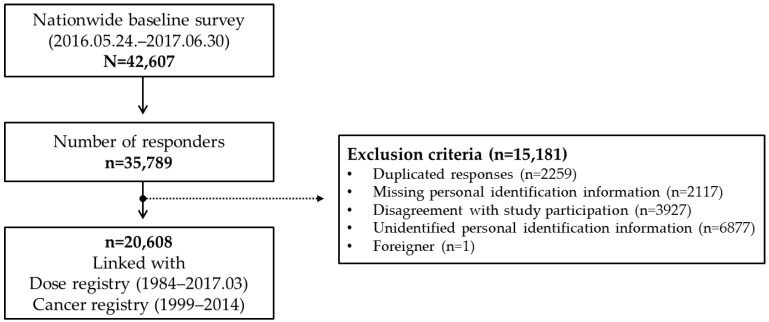
Enrollment of the study population.

**Figure 2 ijerph-17-02328-f002:**
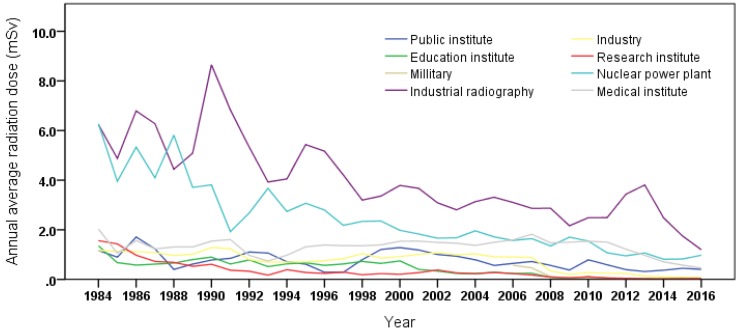
Annual average radiation dose (*H*_p_(10)) by occupation types.

**Figure 3 ijerph-17-02328-f003:**
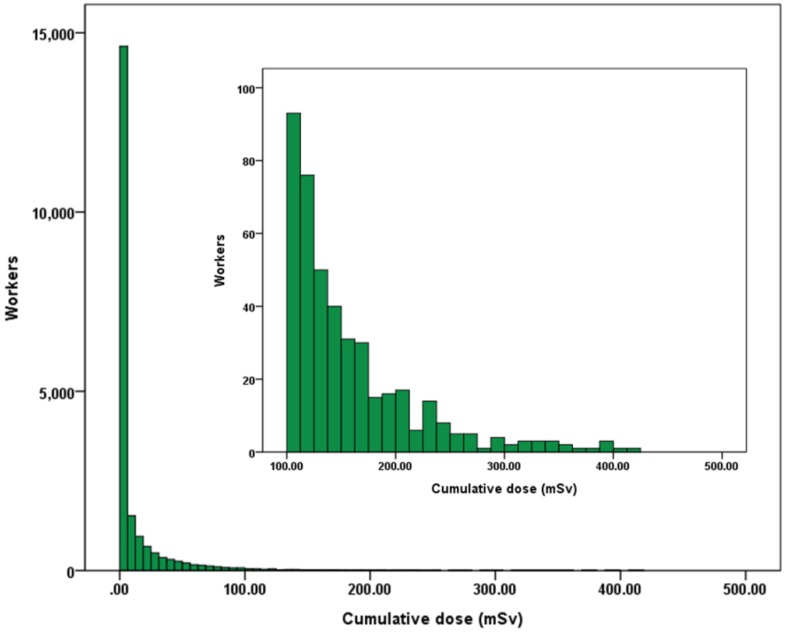
Distribution of cumulative dose in the cohort (1984–the 1st quarter of 2017).

**Table 1 ijerph-17-02328-t001:** Demographic characteristics of the cohort (N = 20,608).

Characteristics	n	(%)
Sex		
Men	17,831	(86.5%)
Women	2777	(13.5%)
Birth year		
~1960	1391	(6.7%)
1961–1970	3449	(16.7%)
1971–1980	5891	(28.6%)
1981~	9877	(47.9%)
Education level		
Less than high school graduation	159	(0.8%)
High school graduation	4539	(22.6%)
College graduation and above	15,402	(76.6%)
Marital status		
Unmarried	7881	(39.2%)
Married/living together	11,982	(59.6%)
Other (divorced, widow, separated)	233	(1.2%)
BMI, kg/m^2^		
Underweight (<18.5)	490	(2.6%)
Normal weight (18.5–24.9)	11,677	(61.2%)
Overweight (25.0–29.9)	6012	(31.5%)
Obese (≥30.0)	913	(4.8%)
Regular exercise		
No	9061	(44.3%)
Yes	11,385	(55.7%)
Smoking status		
Never (non-smoker)	8442	(41.3%)
Ex-smoker	3615	(17.7%)
Yes (smoker)	8361	(40.9%)
Alcohol status		
No	3276	(16.0%)
Yes	17,239	(84.0%)

BMI, body mass index.

**Table 2 ijerph-17-02328-t002:** Occupational characteristics of the cohort (N = 20,608).

Characteristics	n	(%)
Occupation		
Public institute	676	(3.3%)
Education institute	2010	(9.8%)
Military	165	(0.8%)
Industrial radiography	3517	(17.1%)
Industry	3886	(18.9%)
Research institute	1139	(5.5%)
Nuclear power plant	6328	(30.7%)
Medical institute	2887	(14.0%)
Calendar year of hiring		
~1989	1272	(6.2%)
1990–1999	2694	(13.1%)
2000–2009	4569	(22.2%)
2010~	12,073	(58.6%)
Age at the start of radiation work, years		
<20	785	(3.8%)
20–29	12,168	(59.0%)
30–39	5449	(26.4%)
40–49	1565	(7.6%)
≥50	641	(3.1%)
Employment status		
Regular employment	16,450	(82.6%)
Irregular employment (temporary contract)	2885	(14.5%)
Irregular employment (daily contract)	589	(3.0%)
Duration of employment, years		
≤4	9265	(45.0%)
5–9	4310	(20.9%)
10–14	2372	(11.5%)
≥15 years	4661	(22.6%)
Experience of warning for exceeding 5 mSv per quarter		
No	17,857	(90.4%)
Yes	901	(4.6%)
I do not know	987	(5.0%)
Night shifts		
None	10,361	(51.1%)
<1 year	2544	(12.5%)
1–5 years	4215	(20.8%)
>5 years	3152	(15.5%)
Radiation source		
None	2873	(14.6%)
Sealed isotope	5309	(26.9%)
Unsealed isotope	2430	(12.3%)
Radiation-generating device	5559	(28.2%)
Not sure	3539	(18.0%)
Distance from radiation source		
<1 m	3752	(20.0%)
1–3 m	4866	(25.9%)
>3 m	10,149	(54.1%)
While engaged in radiation work, white blood cell counts fell below the normal range		
No	17,922	(90.5%)
Yes	498	(2.5%)
Not sure (or had never had a health examination)	1381	(7.0%)

**Table 3 ijerph-17-02328-t003:** Distribution of radiation doses (*H*_p_(10)) and the number of workers receiving doses >20 mSv per year, 1984–2017.

Report Year	No. of Workers	Annual Collective Dose (man.mSv)	Annual Radiation Dose (mSv [mean ± SD])	No. of Workers >20 mSv Per Year (%)	
1984	417	1876.55	4.5 ± 6.9	25	(6.0)
1985	563	1831.57	3.2 ± 5.5	14	(2.5)
1986	683	3150.87	4.6 ± 7.5	46	(6.7)
1987	870	2923.13	3.4 ± 5.8	25	(2.9)
1988	1085	4210.68	3.9 ± 7.2	64	(5.9)
1989	1213	3428.80	2.8 ± 4.9	20	(1.6)
1990	1348	4567.53	3.4 ± 6.0	42	(3.1)
1991	1491	3494.45	2.3 ± 4.4	21	(1.4)
1992	1690	4145.30	2.5 ± 4.4	19	(1.1)
1993	1763	4557.83	2.6 ± 4.7	24	(1.4)
1994	2041	4554.59	2.2 ± 4.2	18	(0.9)
1995	2457	6528.34	2.7 ± 5.1	47	(1.9)
1996	2725	6803.13	2.5 ± 4.5	32	(1.2)
1997	2890	5958.99	2.1 ± 3.8	14	(0.5)
1998	3140	6199.69	2.0 ± 3.6	7	(0.2)
1999	3502	6887.96	2.0 ± 3.3	6	(0.2)
2000	3696	7041.37	1.9 ± 3.4	14	(0.4)
2001	4008	7156.33	1.8 ± 3.4	9	(0.2)
2002	4212	6922.35	1.6 ± 2.9	10	(0.2)
2003	4463	6950.28	1.6 ± 2.8	8	(0.2)
2004	4925	8413.59	1.7 ± 3.3	15	(0.3)
2005	5302	8732.94	1.6 ± 3.1	17	(0.3)
2006	5812	9113.73	1.6 ± 2.9	14	(0.2)
2007	6447	10,275.9	1.6 ± 2.9	10	(0.2)
2008	6986	9385.69	1.3 ± 3.2	31	(0.4)
2009	7605	9844.92	1.3 ± 3.1	27	(0.4)
2010	8350	11,133.14	1.3 ± 3.3	46	(0.6)
2011	9440	10,743.07	1.1 ± 2.8	30	(0.3)
2012	10,803	13,282.39	1.2 ± 3.4	50	(0.5)
2013	11,965	15,290.65	1.3 ± 3.5	72	(0.6)
2014	13,419	11,881.34	0.9 ± 2.4	18	(0.1)
2015	15,568	10,974.39	0.7 ± 2.2	17	(0.1)
2016	18,861	11,352.00	0.6 ± 1.9	15	(0.1)
2017 **^a^**	18,411	2844.70	0.2 ± 0.7	3	(0.0)
Total	20,608	242,198.04	1.3 ± 3.2	602	(2.9)

**^a^** Radiation doses until the 1st quarter of 2017. SD, standard deviation.

**Table 4 ijerph-17-02328-t004:** Number of observed cases, SIR, and 95% CI among radiation workers in South Korea.

Cancer/Site	Men (n = 16,943)			Women (n = 2584)		
	Obs	SIR (95% CI)		Obs	SIR (95% CI)	
All cancers combined (C00–C96)	189	0.76	(0.66–0.88)	23	0.84	(0.56–1.26)
Oral cavity (C03–C06)	1	1.15	(0.16–8.13)	0	0	(0–99.86)
Salivary gland (C07–C08)	3	2.98	(0.96–9.25)	0	0	(0–42.80)
Stomach (C16)	44	0.95	(0.70–1.27)	1	0.63	(0.09–4.45)
Small intestine (C17)	1	0.84	(0.12–5.94)	0	0	(0–74.90)
Colon (C18)	15	0.93	(0.56–1.54)	0	0	(0–4.68)
Rectum (C19)	10	0.56	(0.30–1.04)	0	0	(0–5.08)
Liver (C22)	7	0.20	(0.10–0.43)	0	0	(0–8.81)
Nose, sinuses, etc. (C30–C31)	2	2.98	(0.74–11.91)	0	0	(0–99.86)
Lung (C33–C34)	4	0.23	(0.09–0.61)	0	0	(0–5.76)
Bone (C40–C41)	1	0.89	(0.13–6.31)	0	0	(0–33.29)
Melanoma of skin (C43)	1	1.41	(0.20–9.98)	0	0	(0–74.90)
Other skin (C44)	1	0.30	(0.04–2.11)	0	0	(0–18.72)
Breast (C50)	0	0	(0–14.97)	9	1.51	(0.79–2.90)
Cervix uteri (C53)	-	-	-	1	0.65	(0.09–4.60)
Uterus, part unspecified (C55)	-	-	-	1	25.86	(3.64–183.60)
Prostate (C61)	5	0.88	(0.37–2.12)	-	-	-
Testis (C62)	2	0.98	(0.24–3.91)	-	-	-
Kidney (C64)	10	1.07	(0.57–1.98)	0	0	(0–13.02)
Bladder (C67)	2	0.46	(0.11–1.82)	0	0	(0–59.91)
Brain, nervous system (C70–C72)	1	0.25	(0.04–1.79)	0	0	(0–11.98)
Thyroid (C73)	72	1.94	(1.54–2.44)	11	0.92	(0.51–1.66)
Adrenal grand (C74)	1	4.29	(0.60–30.43)	0	0	(0–149.79)
Hodgkin lymphoma (C81)	1	1.36	(0.19–9.66)	0	0	(0–42.80)
Non-Hodgkin lymphoma (C82–C86)	2	0.28	(0.07–1.13)	0	0	(0–8.32)
Multiple myeloma (C90)	1	0.86	(0.12–6.10)	0	0	(0–74.90)
Leukemia (C91–C95)	2	0.33	(0.08–1.32)	0	0	(0–7.88)

Obs, observed; SIR, standardized incidence ratio; CI, confidence interval.

**Table 5 ijerph-17-02328-t005:** The SIR and 95% CI for all cancers, thyroid cancer, and all cancers excluding thyroid cancer stratified by occupation and birth year.

Characteristics	n	All Cancers Combined			All Cancers Combined Excluding Thyroid Cancer			Thyroid Cancer		
		Obs	SIR (95% CI)		Obs	SIR (95% CI)		Obs	SIR (95% CI)	
**Occupation**										
Public institute	644	10	1.03	(0.55–1.91)	8	0.98	(0.49–1.96)	2	1.29	(0.32–5.14)
Education institute	1805	10	0.70	(0.38–1.30)	8	0.77	(0.38–1.53)	2	0.51	(0.13–2.05)
Military	164	2	2.25	(0.56–8.99)	0	0	(0–4.68)	2	8.26	(2.07–33.04)
Industrial radiography	3415	12	0.36	(0.20–0.63)	10	0.38	(0.20–0.70)	2	0.28	(0.07–1.13)
Industry	3645	42	0.96	(0.71–1.30)	26	0.75	(0.49–1.05)	16	2.15	(1.32–3.50)
Research institute	1071	17	0.63	(0.39–1.02)	11	0.47	(0.26–0.85)	6	1.75	(0.78–3.89)
Nuclear power plant	6037	93	0.88	(0.72–1.08)	53	0.59	(0.45–0.77)	40	2.68	(1.97–3.65)
Medical institute	2746	26	0.63	(0.43–0.93)	13	0.43	(0.25–0.73)	13	1.23	(0.72–2.12)
**Birth year**										
~1950	103	8	0.45	(0.23–0.91)	8	0.46	(0.23–0.93)	0	0	(0–8.32)
1951–1960	1241	52	0.62	(0.47–0.81)	41	0.52	(0.39–0.71)	11	2.01	(1.11–3.63)
1961–1970	3350	71	0.76	(0.60–0.96)	49	0.62	(0.47–0.83)	22	1.46	(0.96–2.22)
1971–1980	5719	61	1.04	(0.81–1.34)	23	0.58	(0.39–0.88)	38	1.97	(1.44–2.71)
1981–1990	7314	18	0.88	(0.55–1.40)	7	0.59	(0.28–1.24)	11	1.27	(0.71–2.30)
1991~	1800	2	1.85	(0.46–7.40)	1	1.43	(0.20–10.17)	1	2.61	(0.37–18.56)

Obs, observed; SIR, standardized incidence ratio; CI, confidence interval.

**Table 6 ijerph-17-02328-t006:** Relative risks of cancer between the exposed and non-exposed groups (≤0.1 mSv).

Cancer	Exposed Workers (n = 12,065)			Non-Exposed Workers (n = 7462)			Crude RR (95% CI)	Adj^†^ RR (95% CI)
	Obs	SIR (95% CI)		Obs	SIR (95% CI)			
All cancers combined	151	0.69	(0.59–0.81)	61	1.05	(0.82–1.36)	0.64 (0.47–0.86)	0.82 (0.60–1.12)
All cancers combined excluding thyroid cancer	92	0.51	(0.42–0.63)	37	0.79	(0.58–1.10)	0.64 (0.44–0.94)	0.83 (0.56–1.24)
Thyroid cancer	59	1.55	(1.20–2.00)	24	2.17	(1.46–3.24)	0.63 (0.39–1.01)	0.83 (0.49–1.38)

† Adjusted by sex, attained age, birth year, smoke status and duration of employment. Obs, observed; SIR, standardized incidence ratio; CI, confidence interval; Adj, adjusted; RR, relative risk.

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
