# Peer review of "A Cohort Study of Korean Radiation Workers: Baseline Characteristics of Participants"

_ijerph, 2020, doi:10.3390/ijerph17072328_

Round 1
Reviewer 1 Report
I do think this is an interesting study, and given that it is presented as "baseline" there doesn't seem to be anything seriously wrong with its publication. I would, however, appreciate if the authors put more emphasis on the fact that the follow-up time was just about 8 years. Considering that the latency period for most solid cancers is assumed to be 20 - 30 years, this is very short, and no increase in cancer incidence would even be expected on the basis of current knowledge from the Life-Span-Study, for instance.
Abstract, line 12: Is it really „nuclear-related occupations“, or rather „radiation-related occupations“?
Abstract, line 18/19: I think instead of „lower“ and „higher“ it should be „decreased“ and „increased“. Otherwise you would have to say lower or higher than …
Introduction, line 43: Shouldn’t it be „a few studies“?
Fig. 1: Among the „Exclusion criteria“ you have „study population“. That should probably by „participation“.
Materials and Methods, line 72ff: What you collect on a personal badge dosimeter is not the effective dose, but personal dose or personal dose equivalent.
Table 1: From the fact that some study participants „didn’t know“ if they had exceeded 5 mSv per quarter, I guess this is self-reported. So, there is no records of this?
Table 2: Again, should it not be „annual personal dose equivalent“ instead of „annual effective dose“?
Fig. 2: Here you use „Annual average radiation dose“. Should probably be „annual personal dose equivalent“ as well.
Discussion, line 214: Should this not read „one case found among women in the cohort was sufficient to raise the SIR“
Discussion, lines 220-222: „We conducted 602 nationwide surveys that cover approximately 95% of the mandatory radiation safety education for radiation workers for about a year.“ I don’t get the argument here (602 surveys? Or do you mean your survey of 42,607 radiation workers was done in 602 „sub-surveys“ during the mandatory education sessions, and therefore your covered most of the target population? But even so, hoa does that tell you anything about a possible bias with the 20,608 in the cohort?)
Discussion, lines 224-225: „this cohort has limited statistical power at this stage“ If I understand correctly, this is at least partly due to the fact that the usual latency period for solid cancers is considerably more than 8 yeara, maybe 20 or 30. I think this should be mentioned. In fact, I think it should be mentioned in the abstract and conclusions as well. Both only mention very generally „limited follow-up“ so far.
Reviewer 2 Report
Thank you for giving me an opportunity to review this paper. It is a well performed study, but is hard to find direct correlation between radiation exposure to low doses and cancer incidence. Nevertheless, this is a good early attempt, with many papers published on similar concept.
I have a few comments/questions which probably may enhance scientific significance and attention to this paper:
1. In abstract is a sentence: "Further investigations of dose–response associations with long-term follow-up are needed" - please provide explanation to that in discussion part, or remove from abstract.
2. Study population and cohort enrollment part - please provide what type of surveys were performed - i.e. face-to face, via internet etc. The exclusion was made by computer or someone was searching database? Who linked data? I know that appropriate literature is cited, but this part of study is extremly important and should be briefly explained in this part of paper.
3. I'm not sure if non-exposed group should be name like that. It might be confusing for readers, non-exposed mean that somebody is not-exposed, but here it is exposed, but dose is under detection limit. Please consider changing nomenclature and acronyms.
4. Table 1 is detailed, but should be formatted to not be such long.
5. Dose was monitored by personal badge dosimeters - both ring and whole body dosimetres? Please be precisely here.
6. Please check in Table 2: Annual collective dose (man.mSv) - probably should be mean mSv?
7. Please explain briefly in text what are public institutes? What kind of institutions they include?
8. If may happened that someone was working simultaneously in two institutions, how data has been collected in that situation? If that happened, please expalin that briefly in discussion.
Reviewer 3 Report
In their manuscript, Park and co-authors present first results from their cohort study of Korean radiation workers. Results include baseline demographic information, external comparison of cancer incidence rates, and internal comparison using relative risks.
The goal of the study is important and relevant for a wide audience. The methods are sound. The manuscript has a structure that is easy to follow along, and the presentation of background, methods as well as results are clear. However, before publication, a number of issues have to be resolved.
General issues:
Figure 1: By far the largest share (45%) of exclusions are due to unknown causes. This is hard to understand and troubling with respect to selection bias effects. What were the criteria that led to excluding this group?
What information is available on medical radiation exposure (diagnostic / interventional radiology, radiotherapy for benign diseases)? Discuss implications if this information is missing.
Are the SIRs stratified by calendar year group?
Since effective dose is a radiation protection quantity that only partly reflects physical dose, and also incorporates average risk to a reference population, its suitability for individual risk assessment is very limited. For example, effective dose includes radiation detriment for breast cancer which is not applicable for 85% of this cohort. Even though no dose-response analyses are included in this manuscript, this limitation needs to be discussed.
l37: "Effective doses ≤0.1 mSv were recorded as “below recording level” and were considered as a dose of zero for external exposure" - This strategy has severe limitations. Please consult literature on statistical handling of non-detects in environmental studies, e.g, by Helsel. Setting each individual non-detect to 0 can underestimate cumulative dose if the non-detects are > 0 but < 0.1 mSv. Consider sensitivity analyses using a technique like regression on order statistics.
The limit for statistical significance is not defined in the Methods. However, there are multiple tests of the same type carried out, leading to severe multiplicity issues. If the authors choose to report statistical significance, they need to appropriately deal with the multiple testing issue.
In the Discussion, the observed dose distribution should be compared to relevant literature from other dose registries or studies.
Minor issues:
l14: "cumulative dose" - over what time range?
l17: " thyroid cancer was most prevalent (39.2%)" - report number of cases
l18: "(SIR) for all cancers was significantly lower in men" - compared to what, the general population or women? (same for l19)
l32: " studies of radiation workers provide more practical evidence of the health risks of low-dose radiation exposure in our daily lives" - the statement needs some reservation due to healthy worker effect implying a substantial selection effect - as opposed to the LSS study.
l55-l59: This belongs under Discussion
l92: "relative risks" - using what method, Poisson regression taking into account person years?
Table 1: Headings should not be typeset as centered, this makes the table very hard to read.
Table 2: This time-related information is better presented as diagrams with the table in the supplement instead.
l132: "epidemiological studies" - provide citations
l145: "the SIRs ... were lower" - the rates were lower, likewise in l148, l150
Round 2
Reviewer 3 Report
I thank the authors for carefully considering my comments on the initial submission of their manuscript. The issues I raised have been adequately addressed, and I think that the manuscript is now fit for publication.